# Influence of Cross-Shear and Contact Pressure on Wear Mechanisms of PEEK and CFR-PEEK in Total Hip Joint Replacements

Ruimin Shi [1,2], Bukang Wang [1,3], Jiquan Liu [1,3], Zhiwei Yan [1,3] and Lei Dong [2,3,*]

1   Department of Science and Technology Development, Taiyuan Institute of China Coal Technology Engineering Group, Taiyuan 030006, China; shiruimin@nuc.edu.cn (R.S.); bkwang@126.com (B.W.); mkyljq@126.com (J.L.); mkystewart@126.com (Z.Y.)
2   School of Mechanical Engineering, North University of China, Taiyuan 030051, China
3   Department of Test and Simulation, National Engineering Laboratory of Coal Mining Machinery Equipment, Taiyuan 030006, China
*   Correspondence: donglei@nuc.edu.cn

**Abstract:** With the increasing market demand for artificial hip joints, total hip joint replacement has gradually become an effective means of treating a series of hip joint diseases. In order to improve the service life of artificial hip joints, some new artificial hip joint materials, including polyetheretherketone (PEEK) and carbon fiber reinforced polyetheretherketone (CFR-PEEK), have been developed. In this paper, pin-on-plate wear tests under different cross-shear ratios and contact pressures were carried out to study the wear mechanism and worn surface topography of PEEK and CFR-PEEK. The experimental results showed that the wear of PEEK was associated with cross-shear, while CFR-PEEK was not. When the cross-shear ratio was 0.039 and contact pressure was 3.18 MPa, PEEK had poor wear resistance and its wear factor was about eight times that of ultra-high molecular weight polyethylene (UHMWPE). The wear resistance of CFR-PEEK had a significant advantage, since its wear factor was about 30% of that of PEEK. The wear factors of PEEK and CFR-PEEK increased as the contact pressure increased. The arithmetic average of the height amplitude of the surface, Sa, also increased gradually according to the topography of the worn surface. The wear mechanisms of PEEK and CFR-PEEK were scratching, plough cutting, and abrasion Since CFR-PEEK had good wear resistance and insensitivity to cross-shear motion, it is suitable for making artificial hip joints under low contact pressure condition.

**Keywords:** CFR-PEEK; cross-shear; wear; contact pressure; pin-on-plate test; surface topography





## 1. Introduction

In the 1960s, joint replacements provided very good results for many elderly patients with arthritis. Now, young patients are undergoing hip replacement surgery in hopes of being able to engage in more intense physical activity, which will improve their quality of life [1]. Total joint replacement (TJR) arthroplasty was developed to reduce failure rates while accommodating the high levels of activity and longevity of modern patients. Therefore, components must provide durable fixation under high stress, while bearing surfaces need to be resilient and exhibit low wear [2]. The low wear of the contact interface depends on the properties of the contact material and the topography of the contact surface.

Commonly used artificial joint materials can be mainly divided into three categories: metal alloy materials, organic polymer materials and ceramic materials. Among them, cross-linked UHMWPE is the most commonly used as the acetabular component [2,3]; Alison Galvin [4] compared the wear of non-cross-linked and cross-linked UHMWPE and found that high cross-linked UHMWPE could significantly reduce wear when the interface was rough. The increased wear of high cross-linked polyethylene in knee replacement

was caused by an increase in contact pressure. The knee simulator wear test conducted with a 1.7-fold increase in load resulted in a 4.49-fold increase in wear [5]. In the 1990s, researchers investigated the in vitro biocompatibility and stability of different members of the polyaryletherketone (PAEK) family, which showed promise as a replacement for highly cross-linked UHMWPE in bone implant applications [6,7].

Polyetheretherketone (PEEK) has attracted wide attention from researchers in the field of artificial joints because of its good wear, corrosion and fatigue resistance. Using in vivo histological tests of PEEK, it was concluded that it had good biocompatibility [8]. The tribological properties of PEEK and CFR-PEEK are very important in the application of artificial joints [9]. Elliott [10] studied the effects of friction and wear properties of carbon-fiber-reinforced PEEK (CFR-PEEK) by using a pin-on-disk tester. Experimental results showed that CFR-PEEK had lower wear rates than those of PEEK with a sliding speed of 0.18 m/s and a pressure of 1.0 MPa in contact with the stainless steel. Wang A [11] carried out friction and wear experiments on CFR-PEEK and several metal and ceramic materials under simulated hip and knee joint motion. It could be seen from the experimental results that the CFR-PEEK exhibited superior wear resistance compared to the other materials. However, in the simulated knee joint motion, the wear rate of the CFR-PEEK was larger than that of UHMWPE and was independent of contact stress distribution. The wear resistance of CFR-PEEK was lower than that of UHMWPE. Therefore, the CFR-PEEK material was not suitable for knee joint parts [12].

As the molecular orientation of each point on the worn surface of the UHMWPE and PEEK materials changes throughout the motion cycle, the deviation of the sliding vector based on each increment from the sliding direction is not unidirectional. The cross-shear effect refers to the local relative surface motion of the transverse strain hardening direction of the material produced by hip joints during sliding [13]. In the pin-type wear test, the cross-shear effect is reflected in the relative motion form of the gradient of the frictional direction. At the same time, the cross-shear motion is more like the actual motion of the human hip joint. Therefore, cross-shear is an important parameter for measuring the wear performance of artificial hip joint materials [14,15]. Studies have shown that the wear of polyethylene is strongly dependent on the cross-shear effect. Wear can be increased with an increase in cross-shear, and wear depth can be well predicted by using a cross-shear double sine function [16]. Although cross-shear has a great influence on material wear, the mechanism of interaction between them is not fully understood [17].

Currently, in the research of pin-type wear of PEEK and CFR-PEEK with metal or ceramic plates, CFR-PEEK has generally been proven to have the same or better wear performance than UHMWPE under the same conditions [18]. However, there is still a need for further research in areas such as unclear wear mechanisms. Pin-on-disc (PoD) experiments are widely used to quantify and calculate the wear of different material couples for prosthetic hip implant bearings [19]. East RH [20] pointed out that UHMWPE and CFR-PEEK were not suitable combinations because the abrasive properties of carbon fiber could lead to an increase in polyethylene wear. In 2012, 56% of all prosthetic hip implants consisted of metal-on-polyethylene (MoP) bearings [21]. Therefore, this paper studied the wear mechanism of PEEK and CFR-PEEK by using a wear tester for in vitro PoD wear experiments. The variation of wear factors with cross-shear and contact stress were obtained from the tests, where PEEK or CFR-PEEK was used as pins and CoCrMo was used as the plate. The wear mechanisms were obtained by microscopic surface topography analysis and scanning electron microscope (SEM) of the samples after the experiment.

## 2. Materials and Methods

PEEK could be used as an alternative material for artificial hip joints, which has been deeply studied by scholars. The materials selected in the experiment were as follows: PEEK-1000, produced by Ensinger Co., Nufringen, Germany; CFR-PEEK produced by DuPont, Wilmington, DE, USA. CoCrMo of medical grade was used for the counterpart

produced by Shenzhen Hangyu Insulating Plastic Products Co., Ltd. (Shenzhen, China). Its chemical composition and mechanical properties are shown in Table 1.

**Table 1.** Chemical composition and mechanical properties of CoCrMo.

| Material | Co | Cr | Mo | Ni | Fe | Mn | C | Tensile Strength (MPa) | Section Shrinkage (%) | Breaking Elongation (%) |
|---|---|---|---|---|---|---|---|---|---|---|
| CoCrMo | Balanced | 28.08 | 6.13 | 0.1 | 0.1 | 0.24 | 0.099 | 1160 | 29.5 | 31 |

PEEK and CFR-PEEK were processed into cylindrical pins with a contact surface diameter of 8 mm, an outer diameter of 12 mm and a length of 28 mm. In order to reduce the influence of edge burrs on the edge of the cylinder on the test results, a 45° chamfering treatment was performed on the experiment surface. CoCrMo was processed into a 60 mm × 52 mm × 5 mm thin plate, which is shown in Figure 1a. The contact surfaces were subjected to a grinding treatment and the types of sandpaper used were 180#, 400#, 1000#, 3000# and 7000# orderly. Final surface roughness of the pins and plates were both Ra < 50 nm. There was a total of 90 samples in this experiment. Thereafter, the polished samples were ultrasonically cleaned for 30 min using ethanol. The samples were then placed in an electric blast drying box at 60 °C for 2 h. In order to eliminate the error caused by the water absorption of polymer materials in the measurement of wear quality, the method of setting a comparison sample was adopted in the experiment. Before the start of the test, a comparison disc of the same material was placed in the liquid tank, and the same cleaning and drying work was performed on this disc as was performed on the test disc. When the lubricating fluid was changed it was taken out together with the test plate and the same cleaning and drying steps were carried out. The test pin also needed to be equipped with a comparison pin. We measured the height of the pins in the liquid pool during the test, poured the same volume and proportion of lubricating fluid into a small container, put the same material and put the same size comparison pins into it. This method ensured that the same amount of water absorption between the test pins and plate and the comparison ones. Before and after the test, the test piece and the comparison piece were weighed, respectively. The water absorption of the comparison piece was calculated. The weight of the sample before the test was then compared with the one after the test, and the water absorption was measured to obtain the amount of wear in the test piece. The accuracy of the weight balance was 0.01 mg.

The experiment used the multi-functional friction and wear testing machine produced by Rtec Instrument Co., Ltd., San Jose, CA, USA, which has a maximum load of 5000 N and an "X" and "Y" direction bidirectional motor control system. The maximum length of the "X" direction is 150 mm and the maximum allowable speed is 5 mm/s. The maximum length of the "Y" direction is 250 mm and the maximum allowable speed is 40 mm/s. The "Z" direction is controlled by a rotating electrical motor, which has a maximum rotational speed of 2000 rpm, which is shown in Figure 2a.

Cross-shear motion is very similar to the actual motion of a human body, so it has a great influence on the measurement of the wear performance of artificial hip joint materials, which need to be fully considered in the study of wear. The cross-shear ratio (CSR) is defined as the ratio of friction power that is perpendicular to the main molecular orientation (PMO) direction to the total friction power, while the PMO direction is determined as the average direction of the local sliding orbits, or is equivalent to the average direction of friction force. The cross-shear calculation method used in this experiment was based on that of Kang L [22]. The PEEK or CFR-PEEK pins were rotated around their own Z-axis, which was fixed to the upper fixture. The frequency of the rotation speed was 1 Hz ± 0.1 Hz. The CoCrMo plate reciprocated back and forth in the Y-direction with a speed of 1 Hz ± 0.1 Hz, which is shown in Figures 1b and 2b. The reciprocating amplitude of the plates, 20 mm, 28 mm and 38 mm, were measured separately to ensure the consistency of rotation and

reciprocation speed. In different motion states, the artificial joint bears a large stress range [23]. Taking into account the greater load on the hip joints under extreme conditions, the contact pressure in this experiment was set as 3.18 MPa, 6.37 MPa and 9.55 MPa, based on the applied forces of 160 N, 320 N and 480 N, respectively. The experiment conditions and calculated CSR results are shown in Table 2.

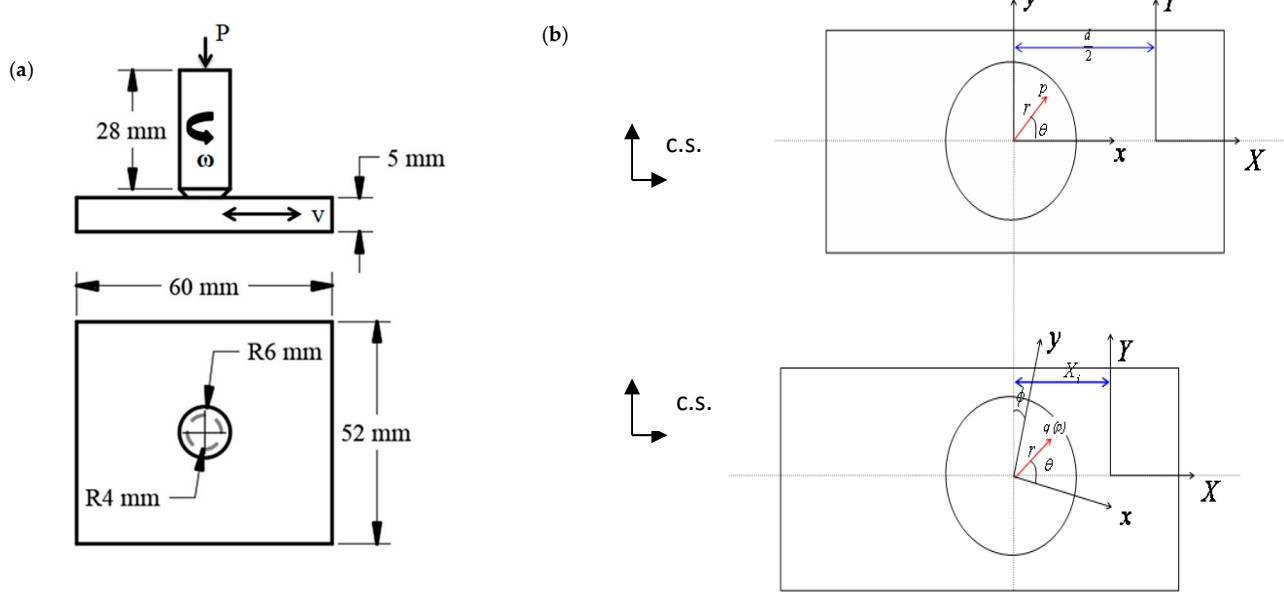

**Figure 1.** Test motion and applied load diagram: (**a**) the pin is loaded and rotating, whereas the plate is sliding; (**b**) the instantaneous position of the pin and plate relative to the starting position at any moment in the coordinate system, c.s.

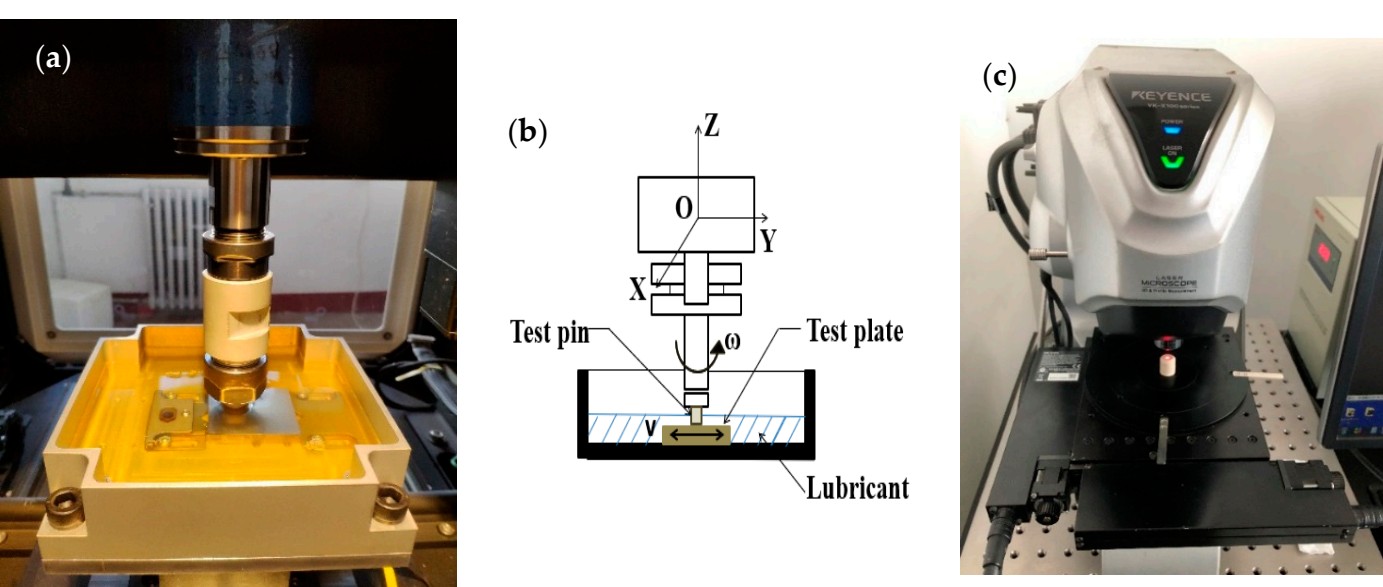

**Figure 2.** Diagram of testing machine: (**a**) friction testing machine (multi-function tribometer MFT-5000, Rtec Instrument Co., Ltd., San Jose, CA, USA); (**b**) schematic diagram of the movement direction and coordinate axis of the wear tester; (**c**) surface topography measurement system (VR-5000, Keyence, Osaka, Japan).

**Table 2.** Experiment conditions.

| Contact Pressure/MPa | 3.18 | | | 6.37 | | | 9.55 | | |
|---|---|---|---|---|---|---|---|---|---|
| Rotating angle of the pin around Z-axis/° | ±20 | ±30 | ±55 | ±20 | ±30 | ±55 | ±20 | ±30 | ±55 |
| Reciprocating amplitude of the plate in Y-direction/mm | 20 | 28 | 38 | 20 | 28 | 38 | 20 | 28 | 38 |
| CSR | 0.039 | 0.087 | 0.254 | 0.039 | 0.087 | 0.254 | 0.039 | 0.087 | 0.254 |

In this experiment, newborn calf serum solution was used to simulate the human body fluid environment. The newborn calf serum was diluted to 25% and supplemented with 0.03% ($v/v$) sodium azide to delay bacterial growth [24]. The newborn calf serum solution that was used in this experiment was produced by Hangzhou Sijiqing Biotechnology Co., Ltd., Hangzhou, China. As the wear was relatively severe in the early stage of the usage period, the test time was 3 h. The test environment temperature was 25 °C. After each experiment, the samples were ultrasonically cleaned in isopropyl alcohol for 10 min. Prior to each weighing, the pin was cleaned and then placed in a controlled temperature and humidity environment for 48 h to stabilize the polymer phase. The sample was further dried in a drying box for at least 30 min and weighed 3 times on the balance by rotating it by a certain angle within 90 min of taking it out of the drying box. The samples were stored in a sealed, dust-free container after tests. The weight was converted to the wear volume and the wear factor, k, was calculated using the Archard wear equation:

$$k = \frac{V}{FS}$$

where $k$ is the wear coefficient (mm$^3$/Nm), $V$ is the volume wear amount (mm$^3$), $F$ is the applied load (N), and $S$ is the sliding distance (m).

As non-contact measurements do not damage the tested surface, this experiment used an optical measurement method to measure PEEK and CFR-PEEK surfaces. The 3D profilometer used in the experiment was the VR series 3D profilometer VR-5000 produced by Keyence (Osaka, Japan). The measuring range was 206 mm × 104 mm, the measuring accuracy was 0.5 μm and the imaging component was a 1 inch and 4-megapixel monochrome complementary metal oxide semiconductor (CMOS) camera (1 inch and 4 million pixels, Keyence, Osaka, Japan), as is shown in Figure 2c. Finally, the wear mechanism of the contact surfaces was studied by using a Hitachi scanning electron microscope, TM4000 Plus. The voltage of the SEM was chosen as 10 kV. Three tests were carried out in each condition and the mean result was recorded.

The research goal of this paper was to reveal the surface topography characteristics and wear mechanism of joint materials PEEK and CFR-PEEK after wear. The significance of this work was to judge whether PEEK and CFR-PEEK are suitable as hip joint materials and predict the wear state of this material through the analysis of the topography and wear mechanism. Therefore, the experiment methods discussed above were designed.

## 3. Results and Discussion

### 3.1. Wear Analysis

Figure 3 show the trend of the mean wear volume of the materials as a function of the experimental cycles under certain shear ratios. Error bars in the figure indicate the magnitude of uncertainty in the measured data. It could be seen that with an increase in the cycle period, the wear volume loss of the materials increased almost linearly, which was basically consistent with the results obtained in the literature [23]. Figure 3a shows the trend of wear volume loss of PEEK with different contact pressures during the experimental cycles. Figure 3b shows the trend of wear volume loss of CFR-PEEK with different contact pressures during the experimental cycles. It could be seen that the contact pressure had a great influence on the wear volume of the materials. The greater the contact pressure was, the greater the amount of wear volume of the materials during the same test cycle.

Compared with traditional PEEK materials, CFR-PEEK had a relatively small amount of wear volume under the same experimental conditions, indicating that the CFR-PEEK materials had good anti-wear properties.

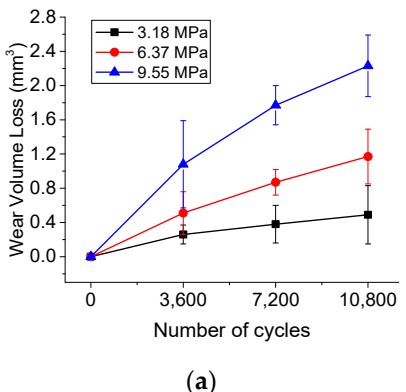
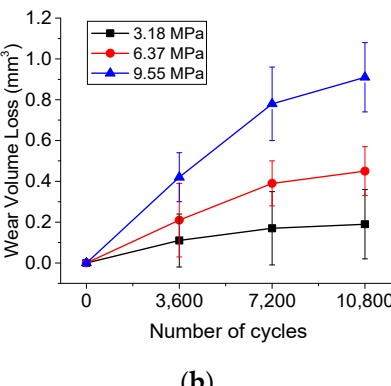

(**a**)  (**b**)

**Figure 3.** Volume loss versus the test time for different contact stresses under a given cross-shear ratio for (**a**) PEEK and (**b**) CFR-PEEK.

Figure 4 shows the trend of the materials' mean wear factor as a function of cross-shear ratio, when the contact pressure was 3.18 MPa. The cross-shear ratio is an important parameter to measure the wear performance of artificial hip joint materials. It can be seen from Figure 4 that in the study of one-way motion (CSR: 0, 20 mm, 0°, 3.18 MPa), the wear factors of PEEK and CFR-PEEK were both of the lowest values. In the study of multidirectional motion (CSR: 0.254, 38 mm, ±55°, 3.18 MPa), PEEK exhibited strong wear behavior associated with cross-shear conditions. The wear factor of PEEK in one-way motion was $(4.75 \pm 0.83) \times 10^{-6}$ mm$^3$/N·m, which was a little larger than that of CFR-PEEK. However, in multi-directional motion with cross-shear ratio of 0.254, the wear factor increased to $(1.27 \pm 0.09) \times 10^{-5}$ mm$^3$/N·m, which was about nine-fold larger than that of CFR-PEEK under the same conditions. The research results were similar to those in [25]. However, the values of the average wear factor of CFR-PEEK materials hardly changed under different cross-shear conditions. The wear factor did not show a regular trend with the increase in the cross-shear ratio. Therefore, the CFR-PEEK did not exhibit wear behavior related to the cross-shear ratio.

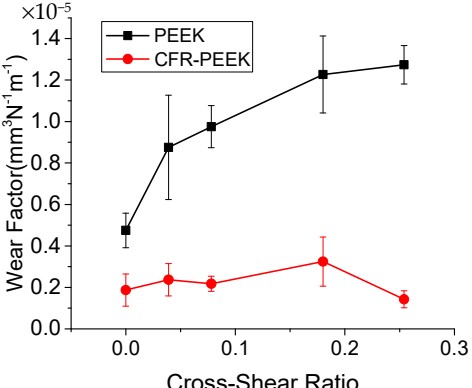

**Figure 4.** Wear factor versus average cross-shear ratio.

The exploration of the performance of new materials has also led to the further development of the research process in the field of artificial hip joints. Different cross-shear ratio conditions were created by several different combinations of displacement and rotation angles to explore the effect of cross-shear ratio conditions on the wear properties of PEEK and CFR-PEEK materials. In comparison with the wear properties of the traditional

joint material UHMWPE, the PEEK materials exhibited wear behavior related to cross-shear ratio conditions. However, changing the cross-shear ratio conditions had little effect on the wear coefficient of CFR-PEEK materials. In addition, the wear factor of PEEK material was higher than CFR-PEEK under all various experimental conditions. Therefore, PEEK material did not have advantages over another metal or polymer in high CS environments. However, PEEK had some other potential uses, for example PEEK is still a good candidate material for low CS bearings, such as rotating platform knee replacements. From the internal properties of the material, the unfilled PEEK material might undergo molecular orientation (strain hardening) in the main direction of motion, which increases the wear performance in this direction while reducing the wear resistance in the vertical plane. The wear factor was also greatly increased under conditions of higher cross-shear ratio. In contrast, randomly oriented carbon fibers in the CFR-PEEK material prevented this reorientation; therefore, the wear properties of the material did not exhibit dependence on the cross-shear ratio. This independence may be beneficial in higher-intensity sport requirements, which are more unfavorable wear conditions. Some studies [20,26] showed that CFR-PEEK materials had great potential for future application of artificial hip joints, which also provided hope for the development of a new generation of high-life artificial hip joints.

Figure 5 shows the trend of the materials' mean wear factors as a function of contact pressure. Three different contact pressure conditions were achieved by applying different loads to the worn contact surfaces and the effect of contact pressure on the wear of PEEK and CFR-PEEK was investigated. Under each experimental condition, the wear factor of PEEK was higher than that of the other material, which also indicated that PEEK had no advantage compared with CFR-PEEK when it was matched with a hard contact surface, such as cobalt chromium alloy. The wear factors of both PEEK and CFR-PEEK tended to increase as the contact pressure increased, but the growth was not significant. This trend was in sharp contrast to the wear performance of UHMWPE, which had been widely demonstrated to have a reduced wear factor as contact pressure increases [27]. Some studies [28,29] showed that the use of ceramic ball heads for articulation with CFR-PEEK hip cups had lower wear. In these designs, the main consideration was that under the "standard" gait conditions, the contact pressure of the implant was relatively low under such conditions, so the wear performance of CFR-PEEK was better than that of the equivalent UHMWPE. However, from the current research and the results of Evans [30] it can be seen that CFR-PEEK had no obvious advantage under high contact pressure conditions, such as edge load and other unfavorable load conditions. Therefore, there is still a need to explore the wear properties of these materials under greater contact pressure conditions, such as climbing stairs or squatting. In addition, Wang [13,14] also explored the effect of the cup angle on the wear of ceramic ball heads and CFR-PEEK hip cups. It was found that when the maximum inclination angle of the cup was 60°, the cup angle had the most significant effect on the wear between the ceramic ball head and the CFR-PEEK hip cup. Therefore, in the practical application of these materials in the field of artificial hip joints, it is also necessary to consider the influence of other more influencing factors on the wear performance.

### 3.2. Surface Topography Analysis

After the experiment, the wear surfaces of the test pieces were scanned using the 3D profilometer to obtain the optical profiles of the wear scar. Figure 6 shows the optical surface topographies of PEEK under a contact pressure of 3.18 MPa and different cross-shear conditions. As shown in Figure 6a, the rough stripe caused by machining in the PEEK wear contact surface in the one-way motion (i.e., CSR = 0) had not been completely removed and the surface was smooth. In the multi-directional motion (see Figure 6b–e), the processing stripes of the PEEK wear surface were completely removed, the material wear level was gradually increased and pits of different shades appeared. Under the experimental conditions of CSR = 0.254, many pits appeared on the wear surface of PEEK and many

scratches were observed at the same time. It could also be seen that under this experimental condition, the level of wear of PEEK was the largest among all experimental conditions.

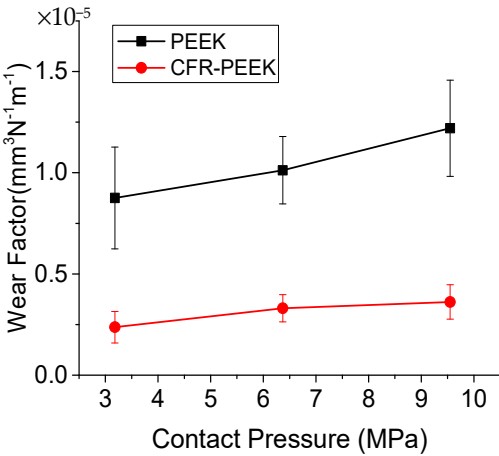

**Figure 5.** Wear factor versus average contact pressure.

Figure 6e,f shows the wear surface profiles of PEEK and CFR-PEEK under CSR = 0.254. It can be seen from Figure 6e that PEEK was worn more severely under multi-directional motion conditions and many peaks or valleys on the wear surface were apparent. At the same time, it could be clearly seen that the wear of CFR-PEEK (Figure 6f) was less than that of the PEEK material and there was no deep peaks or valleys. The wear surface was smoother and the number of pits was also less. This might be because CFR-PEEK had a higher hardness than PEEK and it was less prone to plastic deformation during the grinding process.

Table 3 shows the roughness of the worn surface of PEEK and CFR-PEEK under different contact pressures and cross-shear conditions. Sa was the arithmetic average of the height amplitude of the surface. It is a height parameter for measuring surface roughness in ISO 25178. Overall, with the increase in the contact pressure and cross-shear ratio, the Sa of PEEK and CFR-PEEK both increased. Under the experimental condition of single-directional motion, Sa was about 0.5 µm, which indicated that the wear was relatively slight. Under multi-directional motion, the values of Sa gradually increased. When the condition was CSR = 0.254, the Sa of the worn surface was about 3–4 µm. It could be speculated that due to the adhesion of the new calf serum to the wear debris during the wear process, some of the abrasive debris adhered to the wear surface of the test piece and, as time passed, the wear debris gradually gathered near the contact edge area. This decreased the wear resistance between the contact interfaces and caused the appearance of some pits on the worn surface during the wear process. The Sa of CFR-PEEK was about half that of PEEK when the most severe experiment conditions occurred, which indicated that CFR-PEEK had far better wear resistance than the PEEK material.

Figure 7 shows the optical profiles of the worn surface of the test pieces obtained by scanning the worn surfaces of the test pieces with the 3D profilometer at different contact pressures and cross-shear ratios of 0.039. As shown in Figure 7a, under the condition of low contact pressure, the PEEK contact surface was smooth with only a small number of pits and the level of wear was light. When the contact pressure was increased to 6.37 MPa, the worn surface was rough, with an Sa of about 1 µm. Under the contact pressure of 9.55 MPa, the wear level was more pronounced than in the former case (see Figure 7b). The average Sa of the worn surface was about 1.2 µm and more pits appeared. Figure 7c,d shows the worn surface optical topographies of CFR-PEEK at contact pressures of 6.37 MPa and 9.55 MPa under multi-directional motion conditions with a cross-shear ratio of 0.039. It can be seen that CFR-PEEK had less severe wear scars under these contact pressure conditions and less pits appeared on the worn surface. It can be seen from Table 3 that under a cross-shear ratio of 0.039 the Sa of the PEEK surface was about 1.1 µm and several obvious peaks and

valleys appeared. In comparison, CFR-PEEK also had many peaks and valleys, but the Sa was a little smaller than that of PEEK. Under the same experimental conditions, the worn surface of CFR-PEEK was flat relative to PEEK due to it having fewer deep scars.

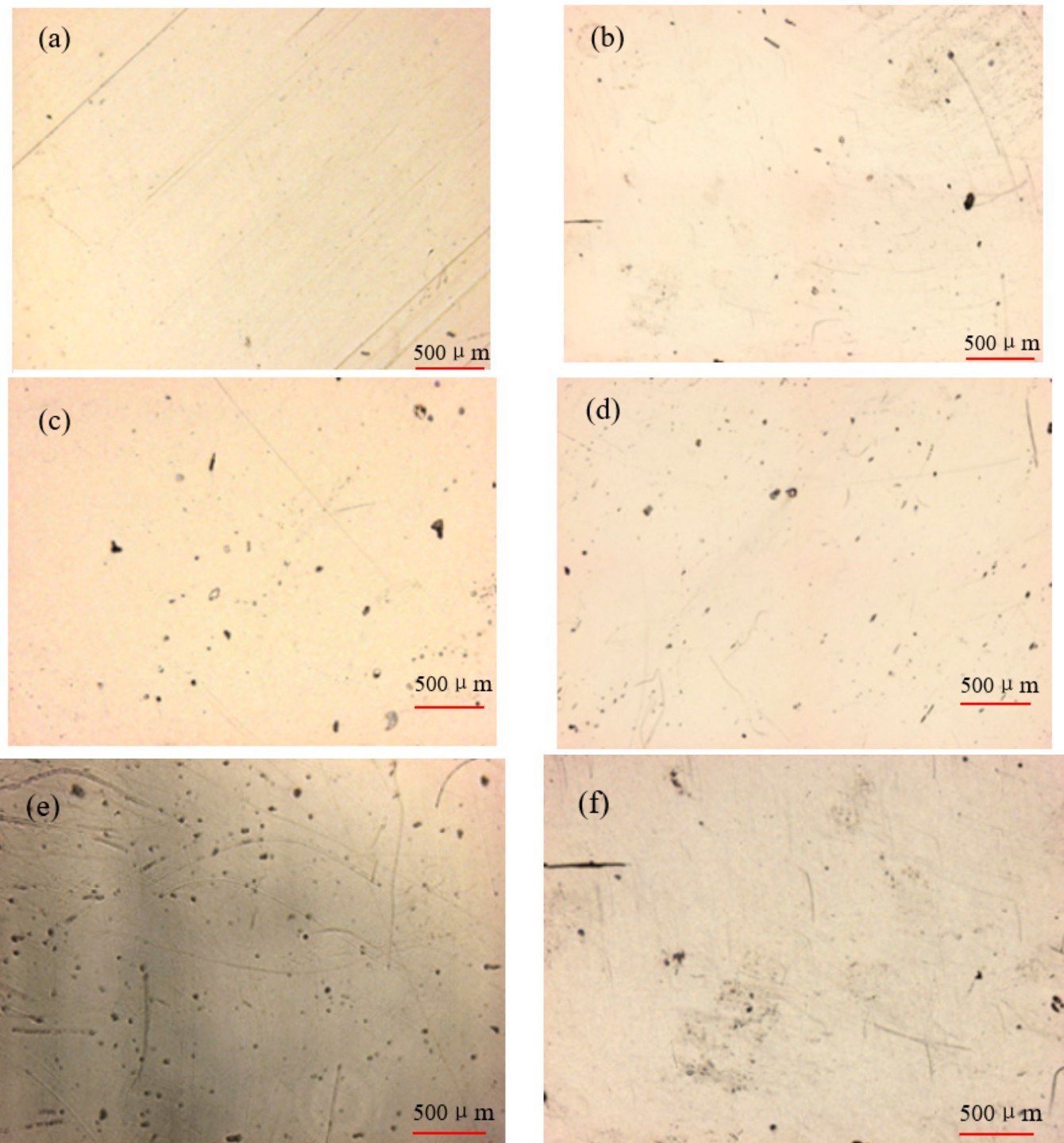

**Figure 6.** Optical surface topographies of PEEK under the same contact pressure of 3.18 MPa but under the following different cross-shear ratio conditions: (**a**) 0; (**b**) 0.039; (**c**) 0.078; (**d**) 0.18; (**e**) 0.254; and (**f**) 0.254 for CFR-PEEK.

**Table 3.** Roughness of the worn surface of PEEK and CFR-PEEK under different cross-shear and contact pressures.

| Cross-Shear Ratio | 0 | | | 0.039 | | | 0.078 | | | 0.18 | | | 0.254 | | |
|---|---|---|---|---|---|---|---|---|---|---|---|---|---|---|---|
| Contact pressure (MPa) | 3.18 | 6.37 | 9.55 | 3.18 | 6.37 | 9.55 | 3.18 | 6.37 | 9.55 | 3.18 | 6.37 | 9.55 | 3.18 | 6.37 | 9.55 |
| Sa of PEEK surface (µm) | 0.49 | 0.52 | 0.77 | 0.81 | 0.98 | 1.21 | 1.52 | 1.74 | 1.98 | 1.82 | 2.14 | 2.35 | 3.17 | 3.55 | 3.97 |
| Sa of CFR-PEEK surface (µm) | 0.28 | 0.49 | 0.66 | 0.51 | 0.79 | 1.09 | 0.88 | 1.05 | 1.29 | 1.13 | 1.36 | 1.38 | 1.19 | 1.44 | 1.68 |

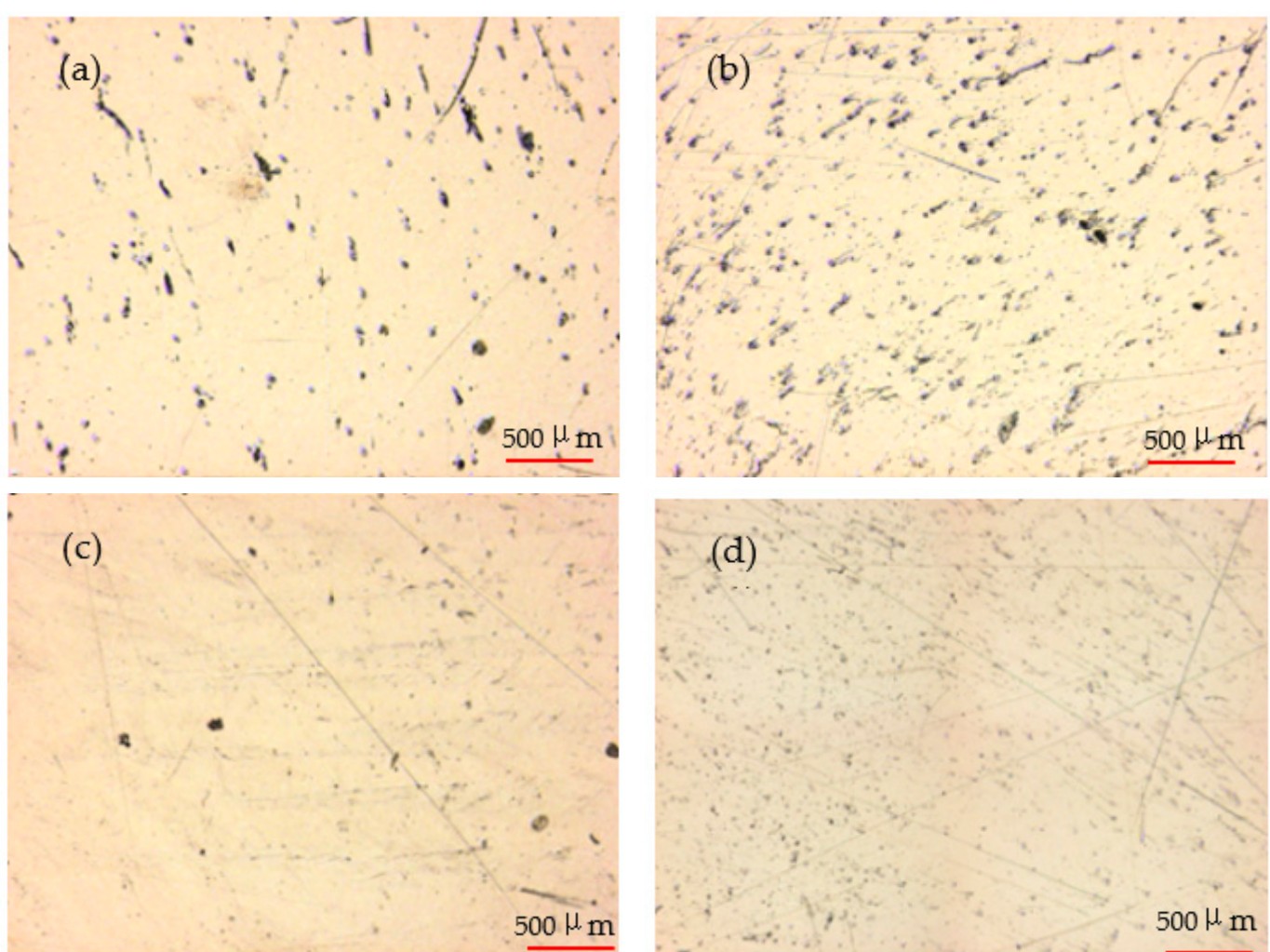

**Figure 7.** Optical surface topographies of PEEK and CFR-PEEK under the same cross-shear ratio at 0.039 but different contact pressures: (**a**) 6.37 MPa for PEEK; (**b**) 9.55 MPa for PEEK; (**c**) 6.37 MPa for CFR-PEEK; (**d**) 9.55 MPa for CFR-PEEK.

### 3.3. Worn Surface Microstructures

The samples under the contact pressures of 3.18 MPa and 6.37 MPa were selected for microstructure characterization. SEM morphologies of worn surfaces of specimens at cross-shear ratios of 0.078 and 0.18 are shown in Figures 8 and 9. It can be seen from the figures that some typical characteristics were observed on the worn surface, such as plough cutting, scratching, abrasion and so on. The worn surface of PEEK was relatively rough (see Figures 8a,b and 9a,b). Scratching and plough cutting were found on the worn surface of PEEK. It indicated that the main wear mechanism of PEEK was the ploughing in the process of cross-shear wear. The SEM morphology of the worn surface of CFR-PEEK is given in Figures 8c,d and 9c,d. Abrasion and some carbide pullouts were found on the

worn surface. The level of wear of CFR-PEEK was lower than PEEK due to the smoother worn surface.

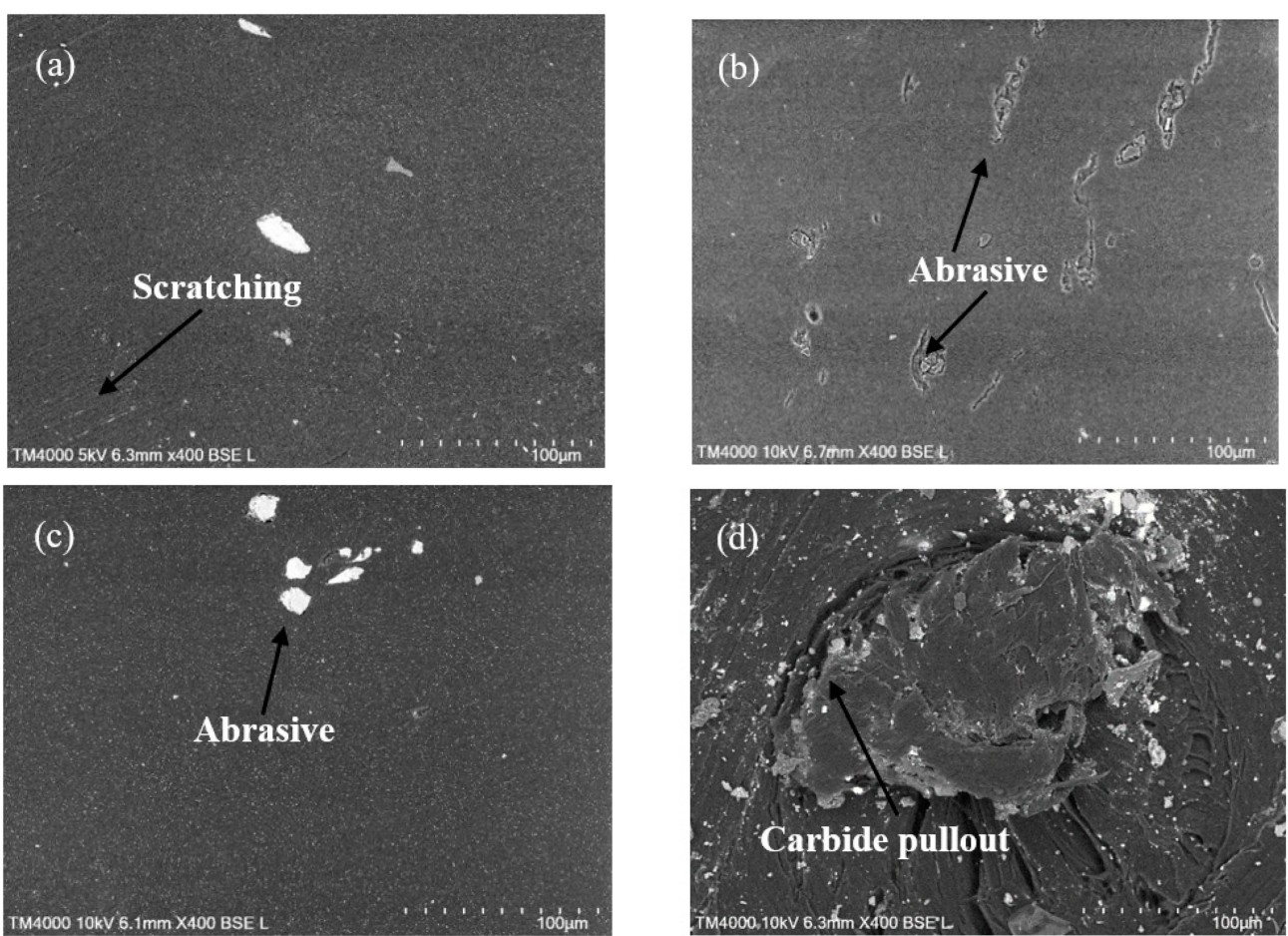

**Figure 8.** SEM morphologies of worn surfaces: (**a**) PEEK with a contact pressure of 3.18 MPa; (**b**) PEEK with a contact pressure of 6.37 MPa; (**c**) CFR-PEEK with a contact pressure of 3.18 MPa; (**d**) CFR-PEEK with a contact pressure of 6.37 MPa under a cross-shear ratio of 0.078.

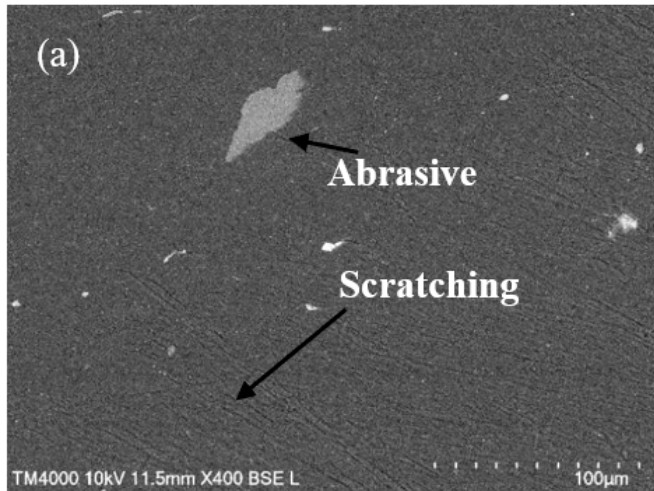
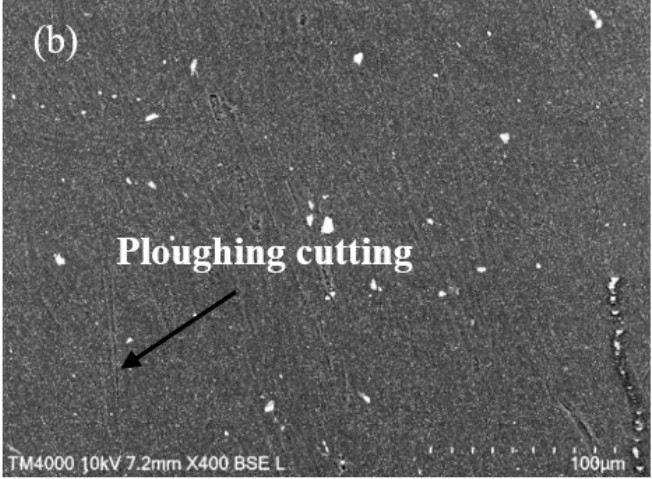

**Figure 9.** *Cont.*

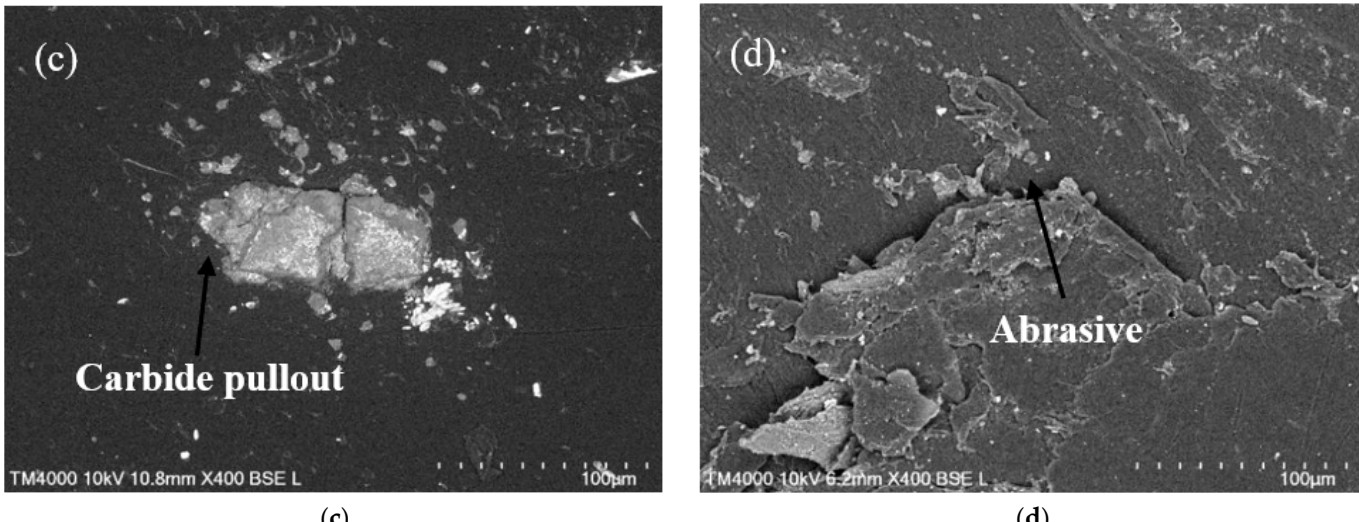

(**c**)                                                                                (**d**)

**Figure 9.** SEM morphologies of worn surfaces: (**a**) PEEK with a contact pressure of 3.18 MPa; (**b**) PEEK with a contact pressure of 6.37 MPa; (**c**) CFR-PEEK with a contact pressure of 3.18 MPa; (**d**) CFR-PEEK with a contact pressure of 6.37 MPa under a cross-shear ratio of 0.18.

As the contact pressure increased, surfaces of PEEK and CFR-PEEK became rougher and had more pits or scratches. Therefore, both materials are not suitable for severe load conditions. CFR-PEEK was, however, not like PEEK, insensitive to the cross-shear motion because there were less differences between the low and high cross-shear ratios, which can be seen in Figures 8d and 9d. As mentioned above, PEEK underwent strain hardening in the main direction of movement, which reduced the wear resistance of the material. So, the wear mechanism of PEEK depended on the cross-shear ratio. In contrast, there were randomly oriented carbon fibers in CFR-PEEK, so the wear performance of the material did not show dependence on the cross-shear ratio. The ploughing of the relatively soft matrix displaced the material binding the reinforcing carbides, thus causing carbide pullout.

## 4. Conclusions

In this paper, the friction pair composed of medical CoCrMo and PEEK or CFR-PEEK was taken as the research object. The wear properties of PEEK and CFR-PEEK as potential materials for artificial hip joint under a series of cross-shear ratios and contact pressures were discussed. The surface topographies and worn surface mechanism of these materials after wear under different testing conditions were studied and analyzed. The main conclusions were as follows:

(1) As the cross-shear ratio increased from 0 to 0.254, the wear factor of PEEK increased from $(4.75 \pm 0.83) \times 10^{-6}$ mm$^3$/N·m to $(1.27 \pm 0.09) \times 10^{-5}$ mm$^3$/N·m, while the wear factor of CFR-PEEK did not show obvious and regular changes with the change in the cross-shear ratio, which basically remained below $2.0 \times 10^{-6}$ mm$^3$/N·m. It could be found that PEEK exhibited a higher wear level than that of CFR-PEEK. Therefore, PEEK is not an ideal replacement for UHMWPE in current artificial hip joint replacement. However, CFR-PEEK exhibited excellent wear performance in the experiment and the wear of CFR-PEEK was not affected by the cross-shear ratio significantly. This showed that CFR-PEEK had great potential in artificial hip joint applications.

(2) From the single-direction motion condition to the multi-directional motion condition, with the development of the cross-shear ratio and contact pressure, the worn surface of PEEK became rougher, the arithmetic average of the height amplitude of the surface Sa also increased from 0.49 to 3.97, and a number of pits gradually appeared on the worn surface. The CFR-PEEK had a lighter degree of wear and its worn surface was smoother than that of PEEK. The Sa of the CFR-PEEK surface was only about

a half of that of PEEK under a cross-shear ratio of 0.254 at a pressure of 3.18 MPa, which indicated that CFR-PEEK was a material with excellent wear resistance in these circumstances.

(3) The wear mechanism of PEEK was scratching, plough cutting and abrasion. Wear mechanism of the CFR-PEEK was abrasion and carbide pullout, which was due to its material properties. In conclusion, CFR-PEEK is suitable as a hip joint material under low load conditions.

**Author Contributions:** Conceptualization, R.S. and L.D.; methodology, Z.Y.; software, J.L.; validation, B.W. and L.D.; formal analysis, R.S.; investigation, L.D.; resources, R.S.; data curation, L.D.; writing—original draft preparation, R.S.; writing—review and editing, L.D.; visualization, R.S.; supervision, B.W.; project administration, B.W.; funding acquisition, R.S. and L.D. All authors have read and agreed to the published version of the manuscript.

**Funding:** This work was supported by the Natural Science Foundation of Shanxi Province (Shanxi Province Department of Finance) (201901D211200, 20210302123065) and Scientific Research Project of Shanxi Tiandi Coal Machinery Equipment Co., Ltd. (Taiyuan Institute of China Coal Technology Engineering Group) (M2021-MS11).

**Institutional Review Board Statement:** Not applicable.

**Informed Consent Statement:** Not applicable.

**Data Availability Statement:** The data presented in this study are available on request from the corresponding author.

**Conflicts of Interest:** No potential conflict of interest was reported by the authors.

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
