# Peer review of "Influence of Cross-Shear and Contact Pressure on Wear Mechanisms of PEEK and CFR-PEEK in Total Hip Joint Replacements"

_lubricants, doi:10.3390/lubricants10050078_

Round 1

Reviewer 1 Report

The paper is generally clear well written.  Some areas for clarification:

Line 87: What was the surface roughness of the PEEK pin?

Line 89: Was the surface roughness of the CoCr plate measured?

Line 129: Provide details of the balance used to measure the mass of the specimens.  What was the precision of the measurements?

Line 133: How many samples were used in the experiments?

Figure captions:  Where there is a graph it needs to be clear if it is a mean value and what the error bars represent.

Figure 6:  Can you also include the images of the specimens before the tests were undertaken.

Table 3: There are 15 columns of numbers – it is not clear what each column represents.

Reviewer 2 Report

General comments:

  • This manuscript needs to be proofread by a native English speaker.
  • The manuscript should either focus on hip or knee joints, not both. The loading and kinematics is substantially different for both, and so are the materials and bearing designs. Especially since the title seems to indicate that one is interested in cross shear, a knee is not a good example since sliding is (almost) unidirectional, in contrast with a hip joint.
  • Introduction section: The authors do not state what the engineering or scientific problem is they try to solve, and why it is significant. Furthermore, the introduction does not provide a focused literature search that documents what others have done, and where a knowledge gap exists. As a result, there is no research objective at the end of the introduction. The authors mention what they describe in the paper, but do not justify it or frame it in terms of significance.
  • Methods section: Rather than providing a complete procedural description of the experiments, it would be useful to explain WHY these procedures were followed and how they play into the objective of this work.
  • The conclusion section should be re-worked. Currently it is a summary. Instead, it should highlight the key takeaways of the research.

Specific comments:

  • P1L29: What do the authors mean by “gradually entering the stage of elderly population”? Why not give quantitative data?
  • P1L30: What is the life expectancy today, and how has it changed? In the US it has actually declined, recently.
  • P1L32: “The development of society and 32 the continuous improvement of related technologies and materials have made artificial joint replacement more and 33 more an effective treatment for joint diseases [1].” This seems to be a real understatement. THR is often considered the “surgery of the century”, see e.g. https://www.thelancet.com/article/S0140-6736(07)60457-7/fulltext
  • P1L34: “Therefore, the demand for artificial joints will also increase greatly”. Please provide quantitative data. Review papers, registries, etc. provide information on this topic.
  • P1L37: “UHMWPE is the most widely used [2,3]”. It’s not today, look at the orthopedic registries. Cross-linked polyethylene (XLPE) is the most commonly used material. Also, one needs to specify which component one evaluates. Here, it refers to the acetabular component. With regards to the femoral head, ceramic (alumina) is the most commonly used material.
  • P1L41: “Artificial joints made of materials 41 such as titanium alloy” Titanium alloys are never used as a bearing material. There is very little titanium wear in prosthetic hip joints.
  • P1L43: “These reactions would cause dissolution 43 near the implanted artificial joint prosthesis, which would further result in the loosening of the well fixed artificial 44 joint prosthesis and affect the quality and life of the artificial joint [6].” This is a really naïve description of osteolysis, and it needs to be strengthened. Furthermore, this is not the dominant failure mechanism in hip implants anymore. What are the other failure mechanisms and how do they compare to ostelolysis?
  • P2L48: “Through the research and evaluation of the tribological properties of PEEK and its com-48 posite materials, it provided a theoretical basis for friction and wear performance in its application in the field of artificial joints [9].” What does this sentence mean?
  • P2L50: “excellent wear resistance” What is excellent wear resistance? Compared to which material? What were the parameters of the experiment?
  • P2L53: “It could be seen from the experimental results that the contact pressure of CFR PEEK exhibited superior wear resistance to other materials” What does this mean? How does contact pressure provide wear resistance?
  • P2L54: “the wear rate of the CFR PEEK was large”. What is large? Compared to what?
  • The introduction should do a more thorough job describing the literature of polyethylene wear for prosthetic joints, especially those that used a PoD experiment. Two papers (and their references) might be useful:

https://www.sciencedirect.com/science/article/pii/S0301679X19300209?via%3Dihub

https://www.sciencedirect.com/science/article/pii/S0301679X18301944?via%3Dihub

  • 1 is not legible, and it is unclear what it attempts to show. If the authors want to show samples, then pictures would be better. If they attempt to show how the experiment was performed, then they should indicate loading and kinematics, and how the specimens interact with each other.
  • 2: is not legible. What is the purpose of the pictures of the instrumentation if the specimens aren’t visible?
  • P4L110: The authors use a combination of a rotating pin with a reciprocating disc. Why? How does this allow tuning the cross-shear and is this cross-shear relevant to the values that occur in prosthetic joints? Is the cross-shear defined in an ISO or ASTM standard?
  • P4L130: It would be helpful to specify that the equation is the Archard wear equation.
  • P4L130: How was wear measured? Gravimetrically? Then conversion to volume for e.g., Fig. 3?
  • 3: Volume loss versus test time is not useful. Test time depends on the frequency of the experiment. Why not have # of cycles on the horizontal axis?
  • 3 and 4: Explain the underlying physics of the results in addition to reporting them. What insights can you share with the reader?
  • Section surface topography analysis: Please expand on this analysis. Why Sa? Why not other metrics? What about the counter surfaces? How do these surfaces look like as a function of time? The way in which one characterizes surface topography depends strongly on the methodology, see e.g., https://www.sciencedirect.com/science/article/pii/S0301679X15004156
  • 7, 8, 9 are not very useful. What can one learn from these individual data points, in terms of general knowledge?

Reviewer 3 Report

The manuscript "Influence of cross-shear and contact pressure on wear mechanism of PEEK and CFR-PEEK for total joint replacement" is interesting and suitable for publication in terms of its central theme. This work may be published after minor corrections as below:

  1. The resolution of figure 1 should be improved to a higher level.
  2. Line 46-47, several recent reviews (doi.org/10.1016/j.actbio.2021.05.009; doi.org/10.1016/j.coche.2021.100687) should be included to support such claim.
  3. How was the pressue calculated in this study?
  4. Line 122, why newborn calf serum was used for the lubricant instead of synovial fluid?
  5. Figure 4-5, part of legend of y-axis is not in the figure.
  6. What was the friction coefficient during the tribological measurements?

Round 2

Reviewer 2 Report

The authors have addressed my concerns, and I recommend the paper to be published. Prior to submitting the final manuscript, I recommend double-checking the list of references for formatting and spelling.

Reviewer 3 Report

I recommend it for publication in Lubricants.